# Changes in the Spatial Structure of Synchronization Connections in EEG During Nocturnal Sleep Apnea

**DOI:** 10.3390/clockssleep7010001

**Published:** 2024-12-31

**Authors:** Maxim Zhuravlev, Anton Kiselev, Anna Orlova, Evgeniy Egorov, Oxana Drapkina, Margarita Simonyan, Evgenia Drozhdeva, Thomas Penzel, Anastasiya Runnova

**Affiliations:** 1Institute of Physics, Saratov State University, Astrahanskaia, 83, Saratov 410012, Russia; zhuravlevmo@gmail.com (M.Z.); evgeniy.n.egorov@gmail.com (E.E.); dr.m-simonyan@yandex.ru (M.S.); drozhdeva.e@bk.ru (E.D.); 2National Medical Research Center for Therapy and Preventive Medicine, Petroverigsky per., 10, Moscow 101000, Russia; kiselev@gnicpm.ru (A.K.); aorlova345@yahoo.com (A.O.); odrapkina@gnicpm.ru (O.D.); 3Laboratory of Open Biosystems and Artificial Intelligence, Saratov State Medical University, Bolshaya Kazachia st., 112, Saratov 410012, Russia; 4Interdisciplinary Sleep Medicine Center, Charite-Universitatsmedizin Berlin, 0117 Berlin, Germany; thomas.penzel@charite.de

**Keywords:** polysomnography, sleep apnea, synchronization, connections, spatial structure

## Abstract

This study involved 72 volunteers divided into two groups according to the apnea–hypopnea index (AHI): AHI>15 episodes per hour (ep/h) (main group, n=39, including 28 men, median AHI 44.15, median age 47), 0≤AHI≤15ep/h (control group, n=33, including 12 men, median AHI 2, median age 28). Each participant underwent polysomnography with a recording of 19 EEG channels. Based on wavelet bicoherence (WB), the magnitude of connectivity between all pairs of EEG channels in six bands was estimated: $Df_{1}$ $\left[0.25;1\right]$, $Df_{2}$ $\left[1;4\right]$, $Df_{3}$ $\left[4;8\right]$, $Df_{4}$ $\left[8;12\right]$, $Df_{5}$ $\left[12;20\right]$, $Df_{6}$ $\left[20;30\right]$ Hz. In all six bands considered, we noted a significant decrease in symmetrical interhemispheric connections in OSA patients. Also, in the main group for slow oscillatory activity $Df_{1}$ and $Df_{2}$, we observe a decrease in connection values in the EEG channels associated with the central interhemispheric sulcus. In addition, patients with AHI>15 show an increase in intrahemispheric connectivity, in particular, forming a left hemisphere high-degree synchronization node (connections PzT3, PzF3, PzFp1) in the $Df_{2}$ band. When considering high-frequency EEG oscillations, connectivity in OSA patients again shows a significant increase within the cerebral hemispheres. The revealed differences in functional connectivity in patients with different levels of AHI are quite stable, remaining when averaging the full nocturnal EEG recording, including both the entire sleep duration and night awakenings. The increase in the number of hypoxia episodes correlates with the violation of the symmetry of interhemispheric functional connections. Maximum absolute values of correlation between the apnea–hypopnea index, AHI, and the WB synchronization strength are observed for the $Df_{2}$ band in symmetrical EEG channels C3C4 (−0.81) and P3P4 (−0.77). The conducted studies demonstrate the possibility of developing diagnostic systems for obstructive sleep apnea syndrome without using signals from the cardiovascular system and respiratory activity.

## 1. Introduction

Obstructive sleep apnea (OSA) is the most common sleep-disordered breathing disorder, characterized by recurrent episodes of cessation of breathing or significant reduction in respiratory flow while maintaining respiratory effort due to collapse of the upper airway. Approximately one billion people worldwide, between 30 and 69 years, have OSA [1].

It is known that OSA is associated with the development of acute and chronic somatic pathology. Thus, airway obstruction leads to an increase in systemic hypoxia, activation of proinflammatory cytokines and activation of the sympatho-adrenal system, which triggers a cascade of pathophysiological mechanisms [2]. In particular, the action of catecholamines causes tachy- and hyperpnea, tachycardia, vasoconstriction and increased blood pressure [3], hyperglycemia and dyslipidemia [4]. These effects in the short term lead to awakening of the patient, rapid reoxygenation of the blood, and alteration of the vascular endothelium by reducing the amount of nitric oxide and increasing the level of endothelin-1 [5], defragmentation, and destruction of the sleep phase ratio [6]. According to the “hypoxia/hypercarbia” theory, repeated episodes of hypoxia and elevated carbon dioxide levels directly damage oxygen-deprived neuronal cells, causing systemic inflammation, oxidative stress, mitochondrial dysfunction, and ultimately ganglion cell apoptosis [7]. Typically, an episode of apnea is accompanied by a decrease in blood oxygen saturation or some change in the ECG rhythm [8].

In addition to short-term effects, an association between OSA and the risk of cardiometabolic disorders has also been shown [9,10], as well as cognitive disorders [11]. Furthermore, the OSA disorder often lacks symptoms and can lead to obesity or cognitive impairment [12,13,14,15,16]. The effect of OSA on the development and progression of chronic kidney disease due to damage to the glomerular apparatus of the kidney is also reported [17].

In clinical studies, cardiorespiratory monitoring methods are usually used to measure the severity of apnea syndrome, without much attention to the electrical activity of the brain. When detecting OSA severity, the clinician’s primary focus is usually on the standard parameter of the apnea–hypopnea index (AHI), which combines multiple characteristics such as the absence of airflow or reduction in airflow accompanied by oxygen desaturations or arousals [18,19]. However, the gold standard for diagnosing obstructive sleep apnea is a full-night polysomnography (PSG) study, which allows for a completely objective description of the patient’s sleep parameters. PSG allows recording of electroencephalographic (EEG) activity of the brain of patients during night sleep. Changes in electrical EEG activity of the brain observed in OSA attract considerable attention both from researchers in the applied fields of neurophysiology, neurology, and neuroscience and in fundamental studies on brain function [20,21,22].

Today, there is no doubt that the occurrence of mild cognitive impairment and emotional imbalance in OSA patients is associated not only with sleep fragmentation and the resulting daytime sleepiness but also with more fundamental reasons correlated with changes in the structural and/or functional activity of the brain [23,24]. Overall, modern neuroimaging studies show structural changes in the brain, in particular, significant changes in white matter areas [25,26] or a decrease in gray matter areas [27,28,29], both of which are now commonly associated with pathological brain aging [30,31,32,33]. Obvious limitations of these studies include the relatively high frequency of comorbid OSA disorders, such as metabolic disorders, hypertension, and obesity in patients with moderate to severe sleep apnea, which themselves are associated with structural brain changes [34,35,36,37] and often relatively small sample sizes due to understandable clinical limitations.

At the same time, the results of neuroimaging studies of structural brain changes in OSA vary significantly. For example, resting-state functional magnetic resonance imaging (rs-fMRI) analysis revealed significant functional connectivity changes in OSA patients, centered on the frontoparietal network, sensorimotor network, and semantic/default mode network, which are closely associated with cognitive impairment [38]. However, a functional MRI study based on solving some problems showed that the insular organization of autonomic responses has a gender bias and is intact only in men with OSA, possibly being impaired only specifically in women [39]. Although, as mentioned above, existing morphological studies mainly report gray matter loss and atrophy in OSA patients [27,40,41,42], there are studies [43,44,45] reporting paradoxical gray matter hypertrophy. Most of the gray matter changes have been described by researchers in the prefrontal lobe, temporal lobe, hippocampus, and cingulate gyrus [27,45,46]. However, it should be noted that the specific locations of changes in the brain are very inconsistent and individualized, their location cannot be fully explained by the severity of the disease, sex/gender, or methodological differences [47].

Thus, the presence of OSA can apparently be considered as a certain factor in pathological aging of the brain. As demonstrated by the results of modern studies, changes in the functional connections of neural networks might precede structural reorganization of the brain [48]. These changes can be revealed by resting-state measurements [49]. Functional connections (FC) assess the dynamic connections between neurophysiological processes, including those based on EEG and MEG measurements. FC assessments, in particular, allow us to distinguish between patients with mild cognitive impairment (MCI) and healthy participants based on the analysis of dysfunction in the cognitive control network (CCN) and dorsal attention network (DAN). Today, it has been shown that sufficient activity and functioning of attentional networks (ventral attention network (VAN), DAN) can apparently compensate for the age-related structural decline in hippocampal volume, allowing episodic memory capabilities to be maintained at a high level [50]. Overall, episodic memory impairment is the most common cognitive-related symptom in patients diagnosed with Alzheimer’s disease (AD) [51]. In mild cognitive impairment (MCI), a subcategory of patients with impaired memory function (amnestic MCI) has been shown to have a high risk of transition to AD [52]. Esposito et al. [53] noted a reduction in anticorrelation activity between the DMN and DAN in MCI patients during rest. Thus, the assessment of both immediate and delayed episodic memory, including additional assessments of attentional networks, may be important for early detection of cognitive decline [54].

In the current situation, studies of the electrical activity of the brain, namely electroencephalography (EEG), are very useful in assessing the preservation of the functional activity of the cerebral cortex. Patients suffering from obstructive sleep apnea demonstrate abnormal spectral patterns on the sleep EEG, characterized by a decrease in power in the low-frequency theta and alpha ranges, as well as high-frequency beta oscillations, which may correlate with indicators of chronic hypoxia [55,56]. At the same time, the analysis of the spectral characteristics of the EEG requires significant averaging and is highly dependent on different phases/stages of sleep or the type of daytime activity of OSA patients, so the EEG analysis often studies the characteristics of EEG functional connectivity, which are more stable in analysis. Studies on cortical effective connectivity during sleep demonstrate significant changes in OSA for rhythmic activity in multiple bands, mainly in delta, alpha, and gamma [57]. It has been repeatedly demonstrated that OSA leads to significant regional changes in EEG functional connectivity interhemispheric interactions [58,59]. In addition, studies have been conducted demonstrating that daytime EEG recordings in OSA patients also have changes in the rhythmic bands’ delta and theta compared with a group of healthy participants [60,61].

According to the results of previous studies, the measure of interhemispheric synchronization is reduced in patients with OSA compared with healthy individuals while experiencing some increase within the cerebral hemispheres [21,22,59]. Rial et al. attribute the impairment of interhemispheric connectivity in OSA patients to the development of asymmetric sleep during oxygen starvation, which may have similar physiological mechanisms with the development of unilateral sleep in marine mammals [62,63]. Zhuravlev et al. demonstrated the detection of increased connectivity of the central and occipital regions for high-frequency EEG activity [59]. Significant changes in functional connectivity were quite stable in groups of apparently healthy participants and OSA patients, maintaining common patterns when comparing both different recording nights and different sleep stages. The maximum variability in connectivity was observed during fast oscillatory processes during REM sleep for the band $\left[30;40\right]$ Hz. In the works of the team that previously studied the data of classical polysomnography of patients’ night sleep, the study of the brain electrical activity was limited to six EEG channels, as shown in Figure 1a. However, such a limited arrangement of EEG channels, although quite sufficient for standard tasks of staging sleep and assessing sleep main parameters, significantly reduced the possibility of studying network structures and synchronization effects arising in the electrical activity of the brain. The purpose of the present study was to expand the understanding of changes in the degree of synchronization and emerging network structures based on studies of night sleep performed with an enlarged EEG record. In this study, EEG signals recorded according to the standard arrangement of “10–20”, as shown in Figure 1b, were processed. This study made it possible to fully study the spatial structure of synchronization connections in the EEG in patients with sleep apnea. To assess the structural properties in functional EEG connectivity during nocturnal sleep, the concept of synchronization in chaotic systems is applied [64,65]. We use the wavelet bicoherence (WB) to estimate the strength of interaction between the brain areas as a measure of synchronization between EEG channels. The WB has proved itself a very powerful instrument to quantify the interactions of various biological systems [66,67,68,69], including brain activity. Gaining a better understanding of the influence of OSA on the pattern of synchronous brain activity has implications for understanding age-related brain changes, including pathological ones such as dementia [70,71]. The present study develops the capabilities of new analytical tools for studying nocturnal EEG recordings and also indicates the possibility of FC EEG analysis as a tool for functional assessment of stress levels in the brain associated with apnea. A distinctive feature of the presented work is the implementation of WB analysis of entire polysomnography records without dividing them into separate fragments according to hypnograms—stages and/or phases.

## 2. Results

For each participant in the study, WB synchronization was calculated between all pairs of EEG channels in the bands $Df_{1}$–$Df_{6}$. The results of the WB synchronization evaluation are presented in the appendix of this manuscript. Then, the obtained estimates were averaged for the main and control groups of test subjects. The average values of the wavelet bicoherence measure, i.e., the spatial structure of synchronization for the two groups of patients under consideration for all pairs of channels, are shown in Figure 2 and Figure 3 for the low- and high-frequency bands, respectively. The average values of WB synchronization between different EEG channels for the main and control groups of study participants are given in the Appendix A. Visual assessment of spatial structures of WB synchronization of EEG in OSA patients and practically healthy participants allows us to say that the observed changes in the network structure are not global but only quantitative and lead to a decrease/increase in the degrees of connections between nodes. The obtained results are in good agreement with previously published works [58,59].

Direct comparison of the control and main groups allows us to observe a decrease in the strength of the average connection between symmetrical EEG channels in patients with significant AHI in all bands under consideration. In addition, a comparison of the presented maps allows us to identify some increase in the level of intrahemispheric connections in both the left and right hemispheres. It is noticeable that when considering an increasing frequency, the average amplitude of the synchronization strength in the spatial structure of connections between different EEG channels grows.

However, both qualitative and quantitative analysis of such spatial matrix maps is quite complex, so spatial difference schemes were constructed, representing the difference between the average measure of synchronization calculated between the control group of practically healthy participants and a similar value calculated for OSA patients. These difference schemes are presented in Figure 4. Here, the brown color corresponds to negative values, i.e., the prevailing values of the strength of connection on the EEG activity in the main group (OSA patients), and the blue color corresponds to positive values, respectively, the prevailing values of the connectivity of the EEG structure in the control group (healthy volunteers).

In the band of the lowest frequencies shown in Figure 4a, the positive values corresponding to intrahemispheric connections are the maximum in absolute value, namely, the connections O1F3, O1Fp1, O1T3, O1F7 located in the left hemisphere and symmetrical for the right hemisphere O2F4, O2Fp2, O2T4, O2F8. Local synchronization features for intrahemispheric connections in such low-frequency bands may be associated with the dynamics of the cardiovascular system and respiratory centers [72,73], but this issue requires further research. The values of connections between hemispheres are reduced to the maximum, namely, those located along the main diagonal—P4P3, C4C3, F4F3, Fp2Fp1. In addition, a significant decrease in the connection values in the leads associated with the central interhemispheric sulcus—CzP3, CzC4, CzC3, CzF4, CzF3, CzFp1, CzT3, CzF7 and FzC4, FzC3, FzF4, FzF3, FzFp2, FzT4, FzF8, FzCz—was quite unexpected. A similar picture of the spatial structure of connections is preserved when considering the frequency interval of 1–4 Hz, becoming brighter, i.e., more pronounced (see Figure 4b). Within the left hemisphere of the brain, for this band of rhythmic activity, the development of a high-degree synchronization node is observed in OSA patients (connections PzT3, PzF3, PzFp1).

A further increase in the ranks of the frequency bands under consideration allows us to observe a certain decrease in the color intensities of the difference matrix, or, in other words, a decrease in the modular values of the observed differences in the estimates of the strengths of connections. The indicated trends, in general, are preserved, demonstrating pronounced stability for different frequency bands. When the frequency exceeds 12 Hz, the oscillatory structure changes significantly. For Figure 4e,f, we observed a marked increase in WB in the OSA patients’ group for the prevailing number of EEG connections considered. This feature in Figure 4f colored the constructed difference scheme almost entirely in brown tones. A decrease in the strength of connections in the OSA group was observed only in symmetrical interhemispheric connections.

Table 1 shows the results of the correlation assessment between the severity of apnea, AHI, and the average characteristics of the WB strength of the connection between EEG channels in frequency bands, $Df_{1}$–$Df_{6}$. The most characteristic pairs of channels are shown, among which maximum absolute values of correlation are observed for $Df_{2}$ band in channels C3C4 (−0.81) and P3P4 (−0.77). The obtained correlation values are reliable according to the *t*-test, p<0.05.

## 3. Discussion

In this study, the functional connection between the brain EEG channels, recorded on people with OSA and healthy subjects, was presented. The calculation of functional connectivity was performed based on wavelet bicoherence. Many degenerative conditions and neuropsychiatric diseases can be caused by or correlated with a failure in the functional networks that make up the brain [74]. This study showed that moderate and severe nocturnal obstructive sleep apnea leads to a significant decrease in interhemispheric symmetrical connections, as well as connections associated with the central sulcus. The maximum differences between OSA patients and healthy participants are observed in the band 1–4 Hz. This difference is also expressed in a significant correlation between the WB connectivity of symmetrical EEG channels and the severity of apnea, AHI. The measure of interhemispheric synchronization between symmetrical EEG leads may potentially be considered a marker of OSA severity in patients.

A recent study Rajeswari and Jagannath showed that the EEG, recorded in the right hemisphere, showed a strong positive correlation association with sleep apnea [75]. Previously, Luo et al. found a significant trend for right hemisphere changes in OSA, which was confirmed by a number of earlier studies using other physical methods, including resting-state functional MRI (rs-fMRI) and positron emission tomography [76]. Horovitz et al. observed a breakdown of connections for the prefrontal cortex in slow-wave sleep [77]. Thus, the cerebral cortex significantly changes the structure of connections, observed, among other things, during the analysis of EEG activity.

In addition, obstructive sleep apnea syndrome has now been linked to structural brain changes in areas associated with memory and Alzheimer’s disease. In particular, a recent study by Martinez showed that OSA is becoming an increasingly recognized risk factor for cognitive decline, linking a significant correlation between higher apnea–hypopnea index and lower functional connectivity through connectivity between the medial prefrontal cortex and bilateral hippocampi, left hippocampus, posterior cingulate cortex, and precuneus [78]. Long, T. et al. described the significance of changes in functional connectivity in the insular subregions and the whole brain of patients receiving successful CPAP-based treatment [79]. Furthermore, there is now some evidence that disturbances in functional EEG connectivity during wakefulness may correlate with specific cognitive problems in OSA patients in the area of memory and attention [80]. However, for direct comparison of the obtained results directly with the activity of various neural networks of the brain, additional studies are required using methods for restoring brain subcortical and cortical sources of electrical activity, for example, based on the independent component analysis method [50] or using convolutional neural network technologies [81].

In our previous study, we performed a mathematical analysis of functional connectivity in six EEG channels of OSA patients and healthy subjects by calculating wavelet bicoherence from nocturnal polysomnograms. In addition to the deterioration of interhemispheric synchronization, it was demonstrated that high-frequency EEG activity includes a compensatory increase in intrahemispheric connectivity and a moderate increase in the connectivity of the central and occipital lobes [59]. Now, this study demonstrates for the first time a powerful change in the associated EEG activity along the central sulcus of the brain of patients and also shows how complex the structure of brain activity behaves in different bands. The obtained results are in good agreement with the theory of the development of “unihemispheric” sleep in patients with apnea, studied in the works [62,63].

The conducted study demonstrates how the spatial structure of EEG connections in OSA patients changes when averaging the entire nighttime polysomnography recording without dividing it into sleep stages/phases. In this case, the averaging periods obviously included both different sleep states and moments of awakening. Thus, the states of wakefulness and sleep are combined, which, nevertheless, demonstrates significant local differences between the control and main OSA groups. Such an approach allows us to hope that the wavelet bicoherence method used is sensitive for diagnosing the features of brain activity in patients both in sleep and wakefulness. The obtained results give us hope that it is possible to detect changes in the structure of connections of the brain’s electrical activity in patients with OSA during a simple daytime recording of brain activity using EEG, i.e., without conducting nighttime sleep monitoring. Obviously, the exclusion of an expensive nighttime study can significantly simplify the detection of this disease and significantly increase the throughput of medical institutions dealing with this issue, since significantly less time will be spent on both recording biophysical signals and processing them per patient. Nevertheless, this assumption requires extensive additional research in this direction.

A limitation of this study is the almost complete absence of cardiovascular problems (hypertension, chronic coronary disease, etc.) in the group of volunteers who do not suffer from OSA (control group, AHI≤15). Such diseases can also have a certain effect on the functional activity of the brain, significantly varying the studied WB synchronization strength. In this regard, further studies should be aimed at assessing the levels of functional connectivity in patients with arterial hypertension and other cardiovascular diseases without OSA. Further development of the method for studying the functional connectivity of brain activity based on the mathematical apparatus of wavelet bicoherence will be aimed at studying the stability of the selected WB characteristics in various groups of patients to clarify its capabilities as an applied diagnostic apparatus for assessing the severity of stress that occurs with apnea syndrome.

It should be noted that the fundamental limitations of the results are also related to the direct method of recording EEG signals, since the activity of neural ensembles in a significant averaged area of the scalp is recorded. In other words, it is not possible to evaluate the behavior of some selective areas of the cerebral cortex. Thus, a direct comparison of functional MRI maps and EEG assessment results of brain activity in different patient groups is of considerable interest.

## 4. Materials and Methods

### 4.1. Materials

Seventy-two subjects participated in our study. The study protocol was approved by the Ethics Committee of the National Medical Research Center for Therapy and Preventive Medicine of the Ministry of Health of Russia, and all experimental procedures were performed in accordance with the ethical standards laid down in the Declaration of Helsinki. All subjects were informed about the experimental procedures in detail and signed standard consent forms. Polysomnographic sleep studies were performed for each participant. The overnight sleep study included the recording of a full “10–20” array EEG (Figure 1b), electrocardiogram, photoplethysmogram, respiratory signal, chin and limb myograms, and oculograms, as shown in Figure 1c. The ECG signal was recorded in standard lead I, according to Einthoven. Respiratory signals were recorded using an oronasal flow temperature sensor and a snore sensor. OCG signals included records of horizontal and vertical eye movements. EEG, ECG, and respiratory function signals were filtered with the bandpass of 0.5–40 Hz and sampled at 100 Hz. EEG signals were acquired with electrodes placed according to the International 10–20 system (specifically, Fp1, Fp2, F3, F4, C3, C4, P3, P4, O1, O2, F7, F8, T3, T4, T5, T6, Fz, Cz, and Pz) using an ear reference. Electrode impedances were maintained below 10 kOm. All PSG records were reviewed by a certified sleep medicine physician for the purpose of nighttime sleep staging.

Patients were divided into two groups depending on the apnea–hypopnea index (AHI), which determines the severity of OSA: AHI>15 episodes per hour (ep/h) (main group, n=39, including 28 men, median AHI 44.15, median age 47), 0≤AHI≤15 ep/h (control group, n=33, including 12 men, median AHI 2, median age 28). Additional patient information is provided in the Table 2. Electroencephalography (EEG) signals were used for further analysis.

### 4.2. Methods

#### 4.2.1. The Wavelet Bicoherence (WB)

We used the wavelet bicoherence to estimate the strength of the connectivity between EEG channels. The wavelet bicoherence has proved to be a very powerful tool for the quantification of the interactions between biomedical signals on various oscillatory scales [66,67,69], including brain activity [68,82,83].

The complex-valued wavelet coefficients $W_{i}\left(f,t_{0}\right)$ for each EEG channel ${\mathrm{EEG}}_{i}\left(t\right)$ was calculated as
(1)$$W_{i}\left(f,t_{0}\right)=\sqrt{f}\int_{t_{0}-4/f}^{t_{0}+4/f}{\mathrm{EEG}}_{i}\left(t\right)p^{*}\left(f,t-t_{0}\right)dt,$$
where i=1,⋯,6 was the number of considered EEG channel, $t_{0}$ was specified the wavelet location on the time axis, “*” denoted the complex conjugate, and $p^{*}\left(f,t\right)$ was the mother wavelet function. We used the standard Morlet wavelet, which was often employed for processing biological signals [69]:(2)$$p\left(f,t-t_{0}\right)=\sqrt{f}pi^{1/4}\exp\left(ß\cdot w_{0}f\left(t-t_{0}\right)\right)\exp\left(\frac{-f^{2}{\left(t-t_{0}\right)}^{2}}{2}\right),$$
where $w_{0}$ was the wavelet scaling parameter and ß was an imaginary unit. The parameter $w_{0}=2pi$ in the continuous wavelet transformation (CWT) provided an optimal time–frequency resolution of the EEG signal [84,85]. To measure the degree of coherence between two EEG signals, ${\mathrm{EEG}}_{i}\left(t\right)$ and ${\mathrm{EEG}}_{j}\left(t\right)$, we used the corresponding complex-valued wavelet coefficients $W_{i}\left(f,t\right)=a_{i}+ßb_{i}$ and $W_{j}\left(f,t\right)=a_{j}+ßb_{j}$.

Wavelet bicoherence, ${\mathrm{WB}}_{ij}\left(f,t\right)$, was estimated based on the mutual wavelet spectrum $W_{i,j}\left(f,t\right)$ of the signals ${\mathrm{EEG}}_{i}\left(t\right)$ and ${\mathrm{EEG}}_{j}\left(t\right)$. Similarly to [68,86], the coefficients Re[WBij(f,t)] and Im[WBij(f,t)], presented as real and imaginary parts of mutual wavelet spectrum, were calculated as
(3)Re[WBij(f,t)]=ai(f,t)aj(f,t)+bi(f,t)bj(f,t)ai2(f,t)+bi2(f,t)aj2(f,t)+bj2(f,t),
and
(4)Im[WBij(f,t)]=bi(f,t)aj(f,t)−ai(f,t)bj(f,t)ai2(f,t)+bi2(f,t)aj2(f,t)+bj2(f,t).

Thus, the synchronization value of two EEG channels, ${\mathrm{EEG}}_{i}\left(t\right)$ and ${\mathrm{EEG}}_{j}\left(t\right)$, at frequency *f* was calculated as follows:(5)WBij(f,t)=(Re[WBij(f,t)])2+(Im[WBij(f,t)])2.With wavelet bicoherence, ${\mathrm{WB}}_{ij}\left(f,t\right)=1$, the signals ${\mathrm{EEG}}_{i}\left(t\right)$ and ${\mathrm{EEG}}_{j}\left(t\right)$ were completely synchronous for time *t* in frequency *f*. Conversely, in the case of zero bicoherence, ${\mathrm{WB}}_{ij}\left(f,t\right)=1$, the signals exhibited a fully asynchronous mode. Accordingly, the value of ${\mathrm{WB}}_{ij}\left(f,t\right)$, changing within the given boundary values [0;1], provided complete information about the connectivity of signals on the time–frequency plane (f,t).

We considered integral bicoherence, calculated from pairs of EEG signals in six frequency bands, $Df_{k}\left[f_{1}^{k};f_{2}^{k}\right]$, Hz, as
(6)$${\mathrm{WB}}_{i,j}^{Df_{k}}\left(t\right)=\frac{1}{Df_{k}}\int_{f_{1}^{k}}^{f_{2}^{k}}{\mathrm{WB}}_{i,j}\left(f,t\right)df,$$
where k=1,…,6 is a number of the considered band $Df_{1}$ [0.25; 1], $Df_{2}$ [1; 4], $Df_{3}$ [4; 8], $Df_{4}$ [8; 12], $Df_{5}$ [12; 20], $Df_{6}$ [20; 30] Hz.

In this case, for each pair of signals, ${\mathrm{EEG}}_{i}\left(t\right)$ and ${\mathrm{EEG}}_{j}\left(t\right)$, six variants of time dependences, ${\mathrm{WB}}_{i,j}^{Df_{k}}\left(t\right)$, could be plotted. For each patient, over the entire duration of the night recording, the mean value of bicoherence $\langle {\mathrm{WB}}_{i,j}^{Df_{k}}\left(t\right)\rangle$ was evaluated for every pair of EEG channels.

#### 4.2.2. Assessment of Spatial WB Connections

To assess the spatial structure, the method of constructing colored matrices was used according to the scheme in Figure 1d. The matrices were formed as intersections of cells designated by the corresponding EEG channels. The main diagonal corresponds to the maximum single connection of the channel signal with itself. The coloring of each cell in the subsequent construction of spatial maps of WB connections was determined according to the value of the calculated average value of WB connection between these channels.

#### 4.2.3. Statistical Analysis

Mean, median, and standard deviation were used in descriptive statistics of collected data. The Mann–Whitney U test for independent samples was performed for the comparison of quantitative data. The results with a *p* value ≤0.001 were assumed to be statistically significant. Statistical analyses were conducted by SPSS version 22.0 software for Windows (IBM, Armonk, NY, USA).

## 5. Conclusions

This study involved 72 volunteers divided into two groups according to the apnea–hypopnea index (AHI): AHI>15 episodes per hour (ep/h) (main group, n=39, including 28 men, median AHI 44.15, median age 47), 0≤AHI≤15 ep/h (control group, n=33, including 12 men, median AHI 2, median age 28). Each participant underwent polysomnography with a recording of 19 EEG channels. Based on wavelet bicoherence (WB), the magnitude of connectivity between all pairs of EEG channels in six bands was estimated: $Df_{1}$ $\left[0.25;1\right]$, $Df_{2}$ $\left[1;4\right]$, $Df_{3}$ $\left[4;8\right]$, $Df_{4}$ $\left[8;12\right]$, $Df_{5}$ $\left[12;20\right]$, $Df_{6}$ $\left[20;30\right]$ Hz. It is shown that the WB changes in the network structure are not global but only quantitative and lead to a decrease/increase in the degrees of connections between nodes, which may indicate some processes of self-organization in the structure of connections of neural ensembles in pathological disorders. In all six frequency ranges considered, we note a significant decrease in symmetrical interhemispheric connections in OSA patients. Also, in the main group for slow oscillatory activity $Df_{1}$ and $Df_{2}$, we observe a decrease in connection values in the EEG channels associated with the central interhemispheric sulcus. In addition, patients with AHI>15 show an increase in intrahemispheric connectivity, in particular, forming in $Df_{2}$ band a left hemisphere high-degree synchronization node (connections PzT3, PzF3, PzFp1). When considering high-frequency EEG oscillations, connectivity in OSA patients again shows a significant increase within the cerebral hemispheres. Maximum absolute values of correlation between the apnea–hypopnea index, AHI, and the WB synchronization strength are observed for $Df_{2}$ band in symmetrical EEG channels C3C4 (−0.81) and P3P4 (−0.77).

Thus, the obtained results demonstrate significant differences in the functional connectivity of the EEG brain activity of patients with different AHI–levels. The differences shown are quite stable, since they are observed when averaging the full polysomnographic record, including both different stages/phases of sleep and nocturnal awakenings. Apparently, nocturnal disturbances of normal respiratory activity lead to the development of the so-called unihemispheric sleep, the degree of which correlates with the severity of obstructive apnea and, apparently, leads to interhemispheric asymmetry in the waking state. Of considerable interest are the prospects for developing diagnostic systems based on the assessment of WB characteristics of the night or even daytime recordings of the patient’s EEG channels. Further directions of the team’s research include expanding the control groups of patients to increase the selectivity of these characteristics, as well as a direct comparison of the obtained results with fMRI characteristics of the brain.

## Figures and Tables

**Figure 1 clockssleep-07-00001-f001:**
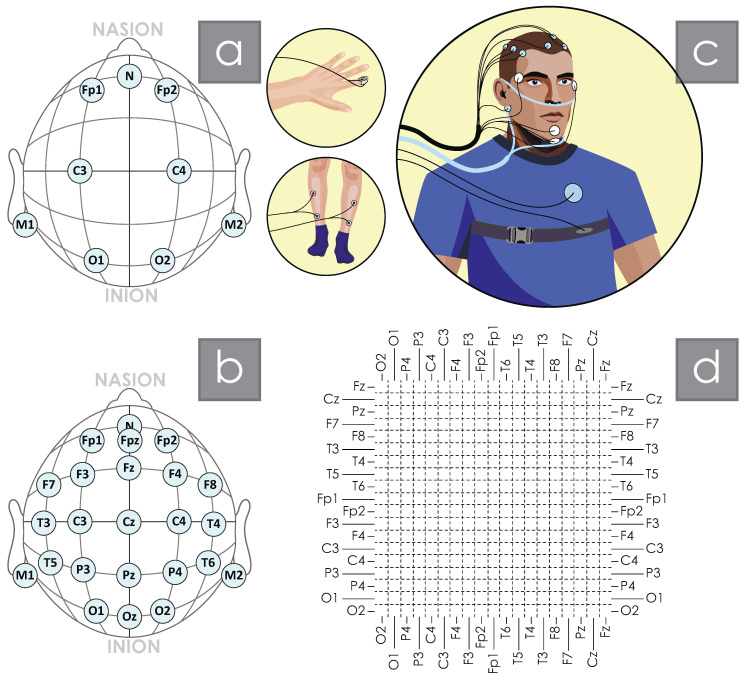
(**a**) diagram of the arrangement of six EEG electrodes used in polysomnographic studies, (**b**) diagram of the arrangement of nineteen EEG electrodes according to “10–20” EEG scheme, (**c**) diagram of the arrangement of polysomnographic equipment during a night study of the patient’s sleep, (**d**) diagram of the presentation of the results of the assessments of synchronization between EEG channels.

**Figure 2 clockssleep-07-00001-f002:**
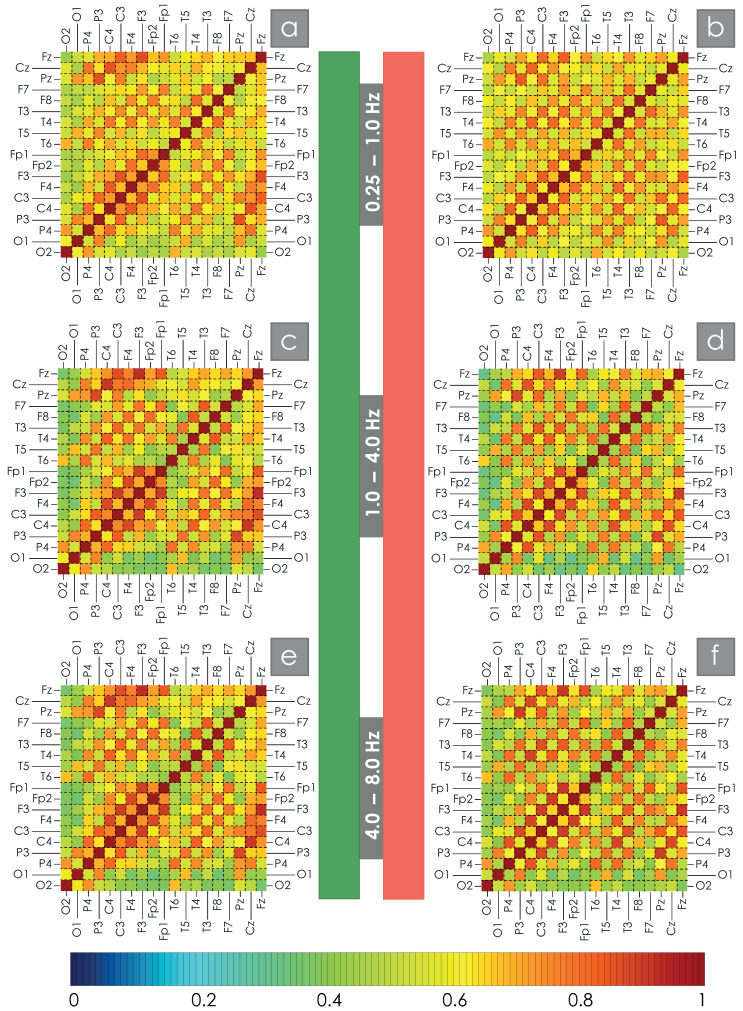
(**a**–**f**) color maps of the spatial structure of synchronization for the frequency bands $Df_{1}\left[0.25;1\right]$, $Df_{2}\left[1;4\right]$, $Df_{3}\left[4;8\right]$ Hz. The left column, marked with a green stripe in the center of the figure, is constructed for the EEG of healthy subjects during night sleep, and the right column (red stripe) corresponds to the OSA data of patients. Below is a color scale demonstrating the correspondence of each color to a certain value of the strength of the average WB connection between each pair of EEG channels.

**Figure 3 clockssleep-07-00001-f003:**
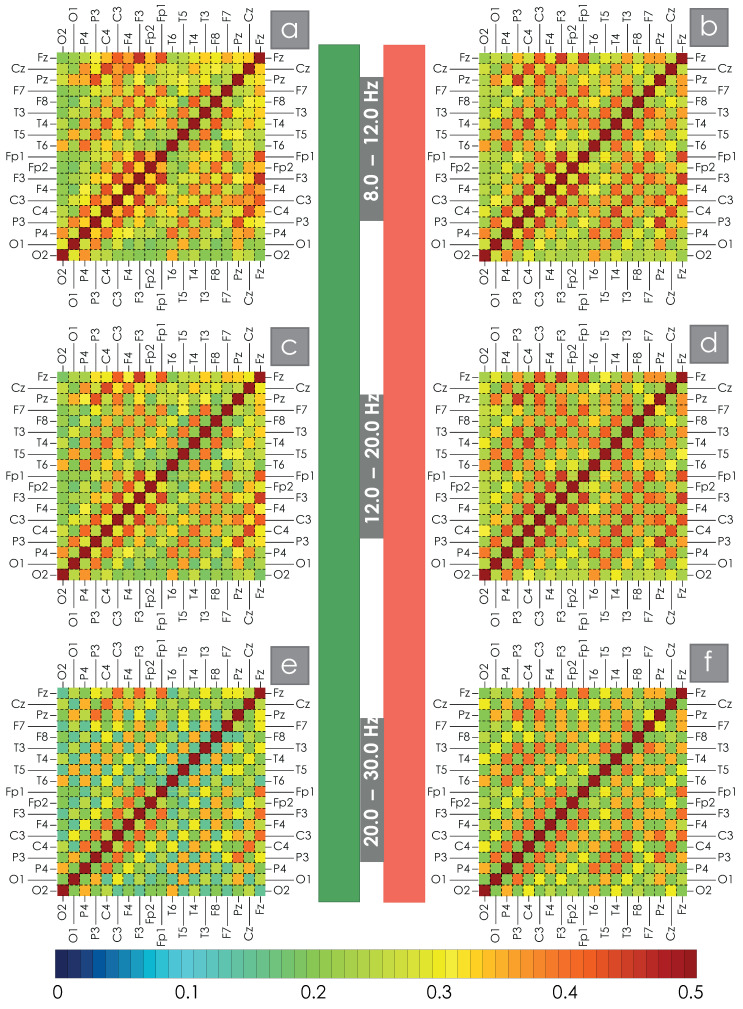
(**a**–**f**) color maps of the spatial structure of synchronization for the bands $Df_{4}\left[8;12\right]$, $Df_{5}\left[12;20\right]$, $Df_{6}\left[20;30\right]$ Hz. The left column, marked with a green stripe in the center of the figure, is constructed for the EEG of healthy subjects during night sleep, and the right column (red stripe) corresponds to the OSA data of patients. Below is a color scale demonstrating the correspondence of each color to a certain value of the strength of the average WB connection between each pair of EEG channels.

**Figure 4 clockssleep-07-00001-f004:**
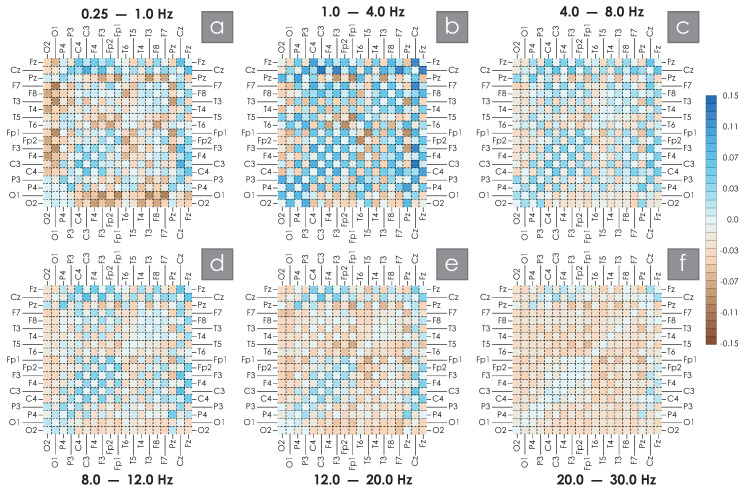
(**a**–**f**) Difference schemes of changes in synchronization measures calculated from EEG signals recorded in groups of healthy subjects and OSA patients in bands $Df_{1}$ [0.25; 1], $Df_{2}$ [1; 4], $Df_{3}$ [4; 8], $Df_{4}$ [8; 12], $Df_{5}$ [12; 20], $Df_{6}$ [20; 30] Hz. On the right is a color scale showing the correspondence between the color designation and the magnitude of the difference between the coupling strengths for each pair of EEG channels.

**Table 1 clockssleep-07-00001-t001:** Correlation characteristics between the apnea–hypopnea index, AHI, and the WB synchronization strength, estimated in EEG channel pairs in different bands $Df_{1}$ [0.25; 1], $Df_{2}$ [1; 4], $Df_{3}$ [4; 8], $Df_{4}$ [8; 12], $Df_{5}$ [12; 20], $Df_{6}$ [20; 30] Hz.

${\mathrm{Df}}_{i}$	C3C4	C3Cz	F3Cz	F3F4	F7Cz	F7F8	O1O2	P3P4
${\mathrm{Df}}_{1}$	−0.73	−0.62	−0.55	−0.57	−0.31	−0.45	−0.32	−0.63
${\mathrm{Df}}_{2}$	−0.81	−0.66	−0.65	−0.70	−0.62	−0.62	−0.62	−0.77
${\mathrm{Df}}_{3}$	−0.71	−0.65	−0.66	−0.65	−0.63	-0.53	−0.53	−0.65
${\mathrm{Df}}_{4}$	−0.71	−0.68	−0.67	−0.65	−0.57	−0.37	−0.36	−0.63
${\mathrm{Df}}_{5}$	−0.70	−0.68	−0.64	−0.60	−0.33	0.11	−0.03	−0.57
${\mathrm{Df}}_{6}$	−0.53	−0.57	−0.50	−0.41	−0.06	0.34	0.14	0.21

**Table 2 clockssleep-07-00001-t002:** Basic physical characteristics of patient groups. The following notations are used: AHI is the apnea–hypopnea index, BMI is the body mass index, SBP is the systolic blood pressure, DBP is the diastolic blood pressure, HR is the heart rate.

	AHI≤15	AHI≥15		AHI≤15	AHI≥15
BMI	21 $\left[19.2;23.6\right]$	32.9 $\left[30.1;38.8\right]$	Hypertension	1 (4%)	21 (53.8%)
SBP	110 $\left[110;120\right]$	130 $\left[120;140\right]$	Chronic coronary disease	0	6 (15.4%)
DBP	70 $\left[70;80\right]$	85 $\left[80;90\right]$	Myocardial infarction	0	3 (7.7%)
HR	76 $\left[73;85\right]$	80 $\left[78;89\right]$	Atrial fibrillation	0	1 (2.6%)

## Data Availability

The datasets generated and analyzed in the course of our study are available on reasonable request from Dr. Anastasiya Runnova (a.e.runnova@gmail.com). The data are not publicly available due to presumed privacy and ethical restrictions.

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
