# Peer review of "Changes in the Spatial Structure of Synchronization Connections in EEG During Nocturnal Sleep Apnea"

_2624-5175, 2024, doi:10.3390/clockssleep7010001_

Round 1
Reviewer 1 Report
Comments and Suggestions for Authors
This is an intriguing and original study. A few things would increase the clinical significance of the paper. 1) Was there a correlation between the oxygen desaturation index and the decreased connectivity or even the correlation between the AHI and the decreased connectivity.
2) Was there any neuroimaging findings on brain MRIs that correlated with the decreased connectivity like reduced brain volume in certain areas or white matter ischemic lesion burden.
Author Response
Thank you very much for taking the time to review this manuscript. Please find the detailed responses below. In manuscript the corresponding revisions and corrections are highlighted in red.
Comments 1: 1) Was there a correlation between the oxygen desaturation index and the decreased connectivity or even the correlation between the AHI and the decreased connectivity.
Respond 1: Thank you for pointing this out. We estimated the correlation between AHI and the WB strength estimated in EEG channel pairs in different bands. The maximum correlation values are observed for the C3C4 channel pair in the D f_2 range.
Comments 2: Was there any neuroimaging findings on brain MRIs that correlated with the decreased connectivity like reduced brain volume in certain areas or white matter ischemic lesion burden.
Respond 2: Unfortunately, our patients generally did not have brain MRIs. However, this is an interesting problem that will be considered as material accumulates in future studies.
Reviewer 2 Report
Comments and Suggestions for Authors
Changes in the spatial structure of synchronization connections in EEG during nocturnal sleep apnea
I have read the manuscript with interest and the topic is novel. However, as a general recommendation, I suggest revising the English language. I did not find the manuscript non-comprehensible but I suggest that is revised by an English native speaker to be improved. You can find my appraisal, as follows:
Introduction: In this section, I have appreciated that you clearly defined the OSA and also reported the epidemiological data. However, OSA and the biochemical mechanism of stress are closely related. This link does not appear in the introduction and I suggest adding a brief paragraph about it. The statement between lines 31-34 is not clear. Please, clarify if you mean that brain activity in OSA patients is studied with tasks. The cognitive alteration observed during resting state fMRI and in resting state networks need to be more specific. Indeed, you described most of the brain networks, but there are specific networks that resulted more affected in their intrinsic connectivity and in their cross-talk (i.e. DAN vs DMN in MCI and different neuropsychiatric diseases). This paragraph needs to be introduced in a better way. I agree that MEG is quite tricky to use, but fNIRS is quite easy to use if compared to HD-EEG systems or 64ch-EEG. I suppose that fNIRS has pros (easy to use) and cons (low spatial resolution, difficulty with dense and curly hair, etc). I suggest removing this sentence. In line 83, you reported a previous study performed by your research group. This needs to be described in a better way since it is part of your aims. Moreover, in the following paragraph (the last of the section), it is present redundant information, previously described that needs to be rewritten or deleted. Similarly, it is not clear the role of WB in deepening the knowledge of OSA and its cerebral correlates (in this case, neuroelectric). So, please, reformulate the hypotheses in light of the previous literature that you also introduced.
2. Materials and Methods: Please change “work” with Study. Please, refer to the general recommendation, as above described. EEG is EEG channel in (1)? The WB was described in a very detailed way. Table 1 needs to be organized in a better way and the acronyms avoided or explained in the caption.
Results: I suggest adding a table (in the supplementary materials if you prefer). I suggest to describe in a better way the results, since they are not clear.
Discussion: The discussion needs to be improved and the results need to be integrated with previously published findings. According to me, this is lacking in this section. I agree that you performed a complex study, but your results have also clinical importance. Please, revise the discussion. Moreover, I advise adding the limitation, the future directions, and the conclusion.
Author Response
Thank you very much for taking the time to review this manuscript. Please find the detailed responses below. We have done a significant amount of work to expand the material presented and hope that this has helped to significantly improve this manuscript. In the manuscript the corresponding revisions and corrections are highlighted in red.
Comments 1: Introduction: In this section, I have appreciated that you clearly defined the OSA and also reported the epidemiological data. However, OSA and the biochemical mechanism of stress are closely related. This link does not appear in the introduction and I suggest adding a brief paragraph about it. The statement between lines 31-34 is not clear. Please, clarify if you mean that brain activity in OSA patients is studied with tasks. The cognitive alteration observed during resting state fMRI and in resting state networks need to be more specific. Indeed, you described most of the brain networks, but there are specific networks that resulted more affected in their intrinsic connectivity and in their cross-talk (i.e. DAN vs DMN in MCI and different neuropsychiatric diseases). This paragraph needs to be introduced in a better way. I agree that MEG is quite tricky to use, but fNIRS is quite easy to use if compared to HD-EEG systems or 64ch-EEG. I suppose that fNIRS has pros (easy to use) and cons (low spatial resolution, difficulty with dense and curly hair, etc). I suggest removing this sentence. In line 83, you reported a previous study performed by your research group. This needs to be described in a better way since it is part of your aims. Moreover, in the following paragraph (the last of the section), it is present redundant information, previously described that needs to be rewritten or deleted. Similarly, it is not clear the role of WB in deepening the knowledge of OSA and its cerebral correlates (in this case, neuroelectric). So, please, reformulate the hypotheses in light of the previous literature that you also introduced.
Respond 1:
Thank you for your careful work with the Introduction, it really allowed us to improve it significantly. First, we added brief information about the connection between OSA and the biochemical mechanism of stress. The added fragments are highlighted in red in the corrected manuscript.
Then, we made a correction to the text on lines 31-34. We implied that in a clinical sleep study, OSA can be diagnosed based only on cardiorespiratory monitoring without monitoring brain activity. However, studying brain activity in this disease is also interesting both in the framework of neurology and neuroscience, and from a fundamental point of view.
Third, we clarified the material on the presence of changes in functional network structures observed in MCI.
Further, we agree with the reviewer's comment and removed the indicated sentence about fNIRS.
We described in more detail the existing studies on the issue of studying FC on EEG brain activity in patients with apnea. Also, based on the background material in the Introduction, they more clearly defined the location of their research
Finally, the last paragraph was removed.
Comments 2: Materials and Methods: Please change “work” with Study. Please, refer to the general recommendation, as above described. EEG is EEG channel in (1)? The WB was described in a very detailed way. Table 1 needs to be organized in a better way and the acronyms avoided or explained in the caption.
Respond 2: Corrected
Yes, EEG is signal registrated in certain EEG channel.
Table 1 is corrected.
Comments 3: Results: I suggest adding a table (in the supplementary materials if you prefer). I suggest to describe in a better way the results, since they are not clear.
Respond 3:
The description of the results has been expanded, as per the reviewer's comment. We have added a section to the results on assessing the correlations between the patients' apnea severity level and WB connectivity between pairs of EEG channels.
In addition, tables with WB FC level assessments have been added to the Appendix to the article.
Comments 4: Discussion: The discussion needs to be improved and the results need to be integrated with previously published findings. According to me, this is lacking in this section. I agree that you performed a complex study, but your results have also clinical importance. Please, revise the discussion. Moreover, I advise adding the limitation, the future directions, and the conclusion.
Respond 4:
We have expanded the Discussion somewhat following the Reviewer's comments, adding limitations of the study as well as future directions for the study.
In addition, a "Conclusion" section has been added to the manuscript:
"The study involved 72 volunteers divided into two groups according to the apnea-hypopnea index~(AHI):~$\textrm{AHI} > 15$ episodes per hour ($ep/h$) (main group, $n=39$, including 28~men, median AHI~44.15, median age~47), $0 \leq \textrm{AHI} \leq 15~ep/h$ (control group, $n=33$, including 12~men, median AHI~2, median age~28). Each participant underwent polysomnography with recording of 19 EEG channels. Based on wavelet bicoherence~(WB), the magnitude of connectivity between all pairs of EEG~channels in six bands was estimated: $D f_1$~$\left[0.25; 1\right]$, $D f_2$~$\left[1; 4\right]$, $D f_3$~$\left[4; 8\right]$, $D f_4$~$\left[8; 12\right]$, $D f_5$~$\left[12; 20\right]$, $D f_6$~$\left[20; 30\right]$~Hz. It is shown that the WB changes in the network structure are not global, but only quantitative and lead to a decrease/increase in the degrees of connections between nodes, which may indicate some processes of self-organization in the structure of connections of neural ensembles in pathological disorders. In all six frequency ranges considered, we note a significant decrease in symmetrical interhemispheric connections in OSA patients. Also in the main group for slow oscillatory activity $D f_1$~and $D f_2$ we observe a decrease in connection values ​​in the EEG channels associated with the central interhemispheric sulcus. In addition, patients with $\textrm{AHI} > 15$ show an increase in intrahemispheric connectivity, in particular, forming in $D f_2$~band a left hemisphere high-degree synchronization node~(connections~PzT3, PzF3, PzFp1). When considering high-frequency EEG oscillations, connectivity in OSA~patients again shows a significant increase within the cerebral hemispheres.
Maximum absolute values ​​of correlation between the apnea-hypopnea index, $AHI$, and the WB synchronization strength are observed for $D f_2$~band in symmetrical EEG channels C3C4~($-0.81$) and P3P4~($-0.77$)."
Round 2
Reviewer 2 Report
Comments and Suggestions for Authors
Thank you for having found my report helpful in improving the manuscript.
In line 278, it is not clear the "Student's criterion". Please, clarify.
Author Response
Comments 1: In line 278, it is not clear the "Student's criterion". Please, clarify.
Respond 1: Corrected:
The obtained correlation values ​​are reliable according to the t - test, p<0.05.